# A dsRNA virus with filamentous viral particles

Hengxia Jia[1,2,3], Kaili Dong[1,2,3], Lingling Zhou[1,2,3], Guoping Wang[1,2,3], Ni Hong[1,2,3], Daohong Jiang[1,2,3] & Wenxing Xu [1,2,3]

Viruses with double-stranded RNA genomes form isometric particles or are capsidless. Here we report a double-stranded RNA virus, *Colletotrichum camelliae* filamentous virus 1 (CcFV-1) isolated from a fungal pathogen, that forms filamentous particles. CcFV-1 has eight genomic double-stranded RNAs, ranging from 990 to 2444 bp, encoding 10 putative open reading frames, of which open reading frame 1 encodes an RNA-dependent RNA polymerase and open reading frame 4 a capsid protein. When inoculated, the naked CcFV-1 double-stranded RNAs are infectious and induce the accumulation of the filamentous particles in vivo. CcFV-1 is phylogenetically related to *Aspergillus fumigatus* tetramycovirus-1 and *Beauveria bassiana* polymycovirus-1, but differs in morphology and in the number of genomic components. CcFV-1 might be an intermediate virus related to truly capsidated viruses, or might represent a distinct encapsidating strategy. In terms of genome and particle architecture, our findings are a significant addition to the knowledge of the virosphere diversity.

[1] State Key Laboratory of Agricultural Microbiology, Wuhan, Hubei 430070, China. [2] College of Plant Science and Technology, Huazhong Agricultural University, Wuhan, Hubei 430070, China. [3] Key Lab of Plant Pathology of Hubei Province, Wuhan, Hubei 430070, China. Correspondence and requests for materials should be addressed to W.X. (email: xuwenxing@mail.hzau.edu.cn)

Viruses infect all cellular organisms including protozoa, bacteria, archaea, invertebrates, vertebrates, algae, plants, and fungi[1]. Their morphotypical peculiarities have been impacted by the environment and the specific nature of the host, which is particularly noticeable in archaeal viruses[2–4]. Viruses that infect plants and fungi display moderate morphotypical diversity, forming bacilliform, icosahedral, or filamentous viral particles (virions), which are closely related with their taxon, evolution, and host[1, 5–8]. Filamentous particles are characteristic of many positive single-stranded RNA ((+)ssRNA)) plant virus families, e.g., *Closteroviridae*, *Potyviridae*, *Alphaflexiviridae*, *Betaflexiviridae*, and *Gammaflexiviridae*, but this is not the case in double-stranded RNA (dsRNA) viruses, even in those isolated from other organisms including protozoa and animals[1]. Since previous studies have revealed that dsRNA viruses may have repeatedly originated from distinct supergroups of (+)RNA viruses[8, 9], it is striking that such filamentous architecture has not been observed in any dsRNA virus, thus keeping the evolutionary relationships between the morphologies of (+)ssRNA and dsRNA viruses enigmatic.

DsRNA viruses are a diverse group that infect a wide range of hosts, from bacteria to eukaryotes including fungi, protozoa, plants, and animals[1]. Most dsRNA viruses present isometric particles, including members of the families *Totiviridae* (non-segmented genome, 4.6–7.0 kbp), *Partitiviridae* (two or three genomic segments, 1.4–2.3 kbp), *Chrysoviridae* (three to five genomic segments, 2.4–3.6 kbp), *Reoviridae* (10–12 genomic segments, 0.7–5.0 kbp), *Megabirnaviridae* (two genomic segments, 7.0–9.0 kbp), *Quadriviridae* (four genomic segments, 3.7–4.9 kbp), and the proposed family "Alternaviridae" (four genomic segments, 1.4–3.6 kbp)[10, 11]. Some dsRNA viruses do not form virions but are associated with or enveloped by colloidal proteinaceous components, as observed recently for the mycovirus *Aspergillus fumigatus* tetramycovirus-1 (AfuTmV-1) from the human pathogenic *A. fumigatus*[9], and for the mycovirus *Beauveria bassiana* polymycovirus-1 (BbPmV-1) from insect pathogenic *B. bassiana*[12].

Here we report the isolation and characterization of a dsRNA virus from a strain of *Colletotrichum camelliae* Massee (LT-3-1) infecting tea (*Camellia sinensis* (L.) O. Kuntze) in China. With its

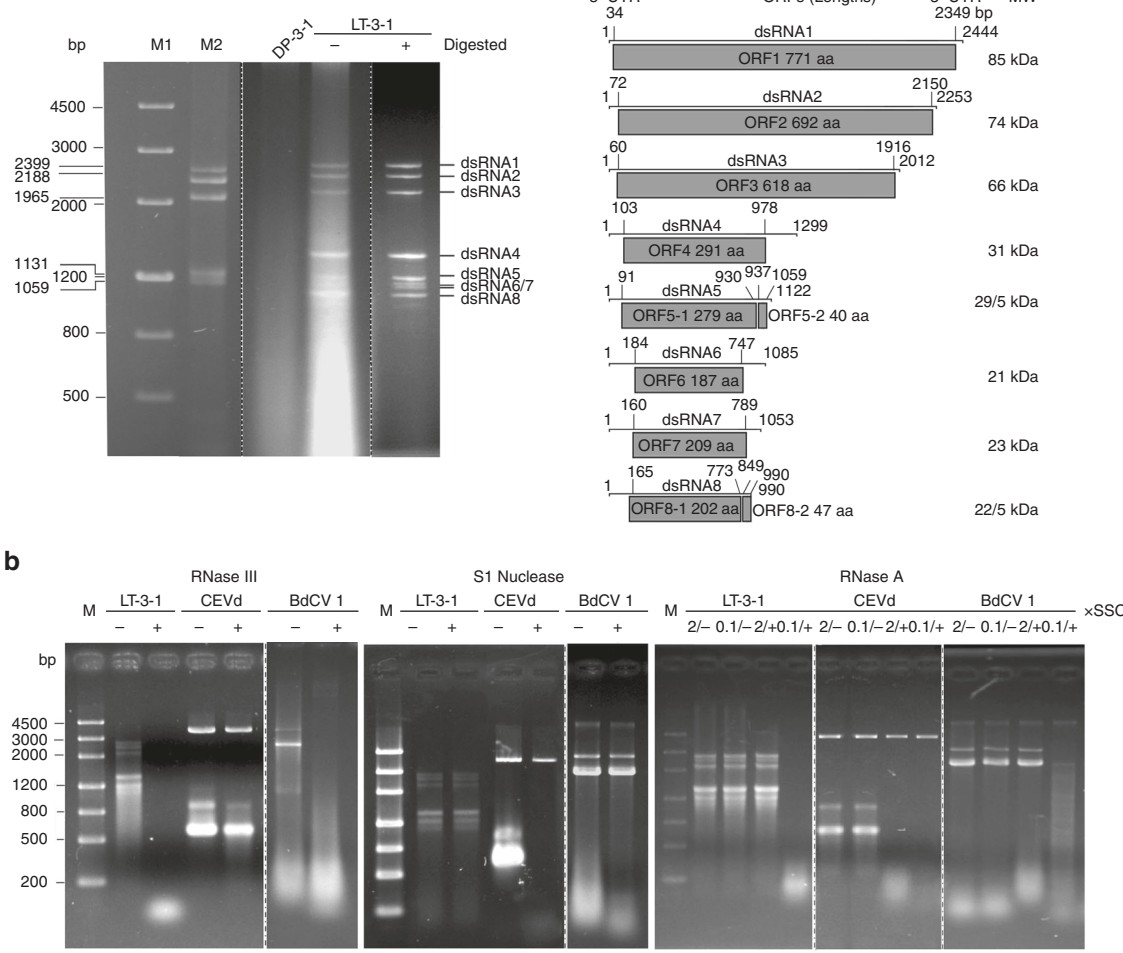

**Fig. 1** Electrophoresis analysis, enzyme treatment, and genomic characteristics and organization of the eight dsRNA segments extracted from mycelia of *Colletotrichum camelliae* strain LT-3-1. **a** Electrophoretic profiles on a 1.2% agarose gel of dsRNA preparations from strain LT-3-1 before (−) and after (+) digestion with DNase I and S1 nuclease, and from strain DP-3-1 after digestion with both enzymes. Nucleic acid sizes are indicated beside the gels. **b** Electrophoresis analysis of enzyme-treated nucleic acid samples on 1.2% agarose gels. The samples were treated with RNase III, S1 nuclease and RNase A (in 2× and 0.1× SSC), respectively. "−" and "+" refer to incubated in the reaction buffer without and with the enzyme, respectively. CEVd and BdCV 1, ssRNA transcripts from dimeric cDNAs of citrus exocortis viroid (*CEVd*), and dsRNA extracts from mycelia of *Botryosphaeria dothidea chrysovirus* 1 (*BdCV 1*), respectively. The *upper bands* on the lane of CEVd sample correspond to the remnant plasmid used for transcription, and the *lower bands* to the transcripts (two bands due to conformation difference). **c** Genomic organization of dsRNAs 1–8 showing putative open reading frames (*ORFs*) and untranslated regions (*UTRs*). The *dot line* refers to the separation of the both gels migrated in separate lanes with treatments in parallel

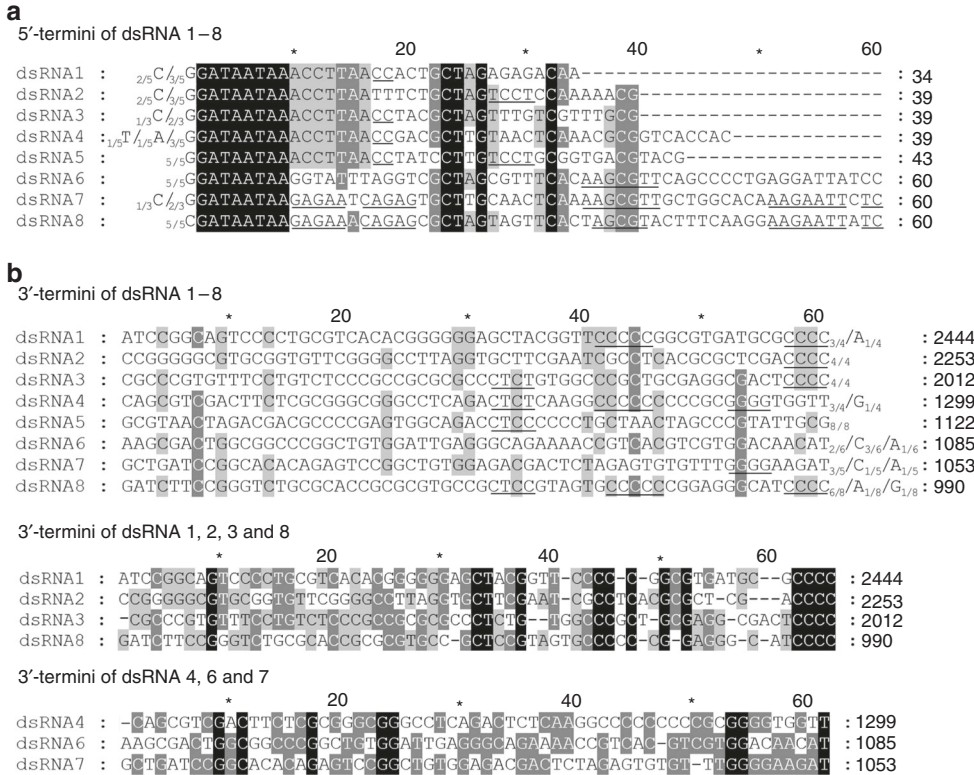

**Fig. 2** Multiple alignments of the terminal regions of the coding strands of dsRNAs 1–8. **a**, **b** Conserved sequences of the 5′- and 3′-termini of the dsRNAs, respectively. *Black, gray,* and *light gray backgrounds* denote a nucleotide identity of no less than 100%, 80%, and 60%, respectively. The terminal nucleotides together with their frequencies in the RACE experiments are indicated adjacent to the strand ends

flexuous and elongated viral particles containing a dsRNA genome of eight fragments, this virus displays molecular and structural features that have, to the best of our knowledge, not been previously observed in dsRNA viruses. These features provide insights into the evolution of this group of viruses.

## Results

**A complex pattern of dsRNAs in *C. Camelliae* strain LT-3-1.** Nucleic acid preparations enriched in dsRNA were obtained from mycelia of *C. camelliae* strain LT-3-1 and analyzed by agarose gel electrophoresis. A complex pattern of eight bands was detected in LT-3-1 preparations before and after digestion with DNase I or S1 nuclease (Fig. 1a). Assuming that these bands were generated by dsRNAs, their corresponding sizes were between 2500 and 900 bp as estimated by agarose gel electrophoresis using both dsDNA and dsRNA markers (Fig. 1a). These RNAs were not detected in a typical strain-like DP-3-1 (Fig. 1a).

The dsRNA nature of the eight observed bands was assessed by treatments with RNase III, S1 nuclease, or RNase A (in 2× and 0.1× SSC), together with an ssRNA control (in vitro dimeric transcripts of citrus exocortis viroid (CEVd)) and dsRNAs from a dsRNA mycovirus (*Botryosphaeria dothidea* chrysovirus 1, BdCV 1). The RNAs extracted from strain LT-3-1 together with BdCV 1 dsRNAs were digested into ~20 bp-sized fragments by RNase III and degraded by RNase A in 0.1× SSC, but they resisted digestion by S1 nuclease and RNase A in 2× SSC. In sharp contrast, CEVd transcripts were completely degraded by S1 nuclease and by RNase A under both ionic conditions, but resisted digestion by RNase III (Fig. 1b). These data strongly support that the RNAs extracted from strain LT-3-1 were indeed dsRNAs (hereafter termed dsRNAs 1–8 according to their decreasing size).

**Complete sequence and genomic organization of dsRNAs 1–8.** The sequences of the full-length complimentary DNAs (cDNA) of dsRNAs 1–8 were determined by assembling partial cDNAs amplified from each individually purified dsRNAs using real-time (RT) PCR with tagged random primers and RACE[13, 14]. Regarding the shortest dsRNAs 7 and 8, following ligation of a 3′-closed adaptor (46 nt) to the 3′ end of each RNA strand, they reverse transcribed using a primer complementary to the adaptor (positions 46–26), PCR-amplified with a second primer complementary to the adaptor (positions 34–17), and reamplified by nested PCR with a third primer complementary to the adaptor (positions 17–1). Cloning and sequencing the amplified products confirmed their full-length size. The sizes obtained for dsRNAs 1–8 were 2444, 2253, 2012, 1299, 1122, 1085, 1053 and 990 bp, respectively; the corresponding sequences have been deposited in GenBank under accession numbers KX778766-KX778773. Each dsRNA contains one (dsRNAs 1–4 and dsRNAs 6 and 7) or two (dsRNAs 5 and 8) putative open reading frames (ORFs) on one of the strands; the ORFs were named according to the dsRNA on which they are located, e.g., ORF 5-1 is the first ORF on dsRNA 5 (Fig. 1c). BLASTn searches revealed that dsRNA1 sequence shared low similarities (3.0E$^{-44}$–2.0E$^{-10}$; 25–41% coverage; 68–82% identity) to the dsRNA1 of four mycoviruses including two uncharacterized ones: *Alternaria tenuissima* virus (ATV, accession nos. KP067914), *Cladosporium cladosporioides* virus 1 (CcV-1, KJ787686), AfuTmV-1, HG975302, and *Botryosphaeria dothidea* virus 1 (BdRV1, KP245734). However, the remaining RNAs (2–8) shared no detectable similarities with viral RNA sequences.

The 5′-untranslated regions (5′-UTRs) of the coding strand of dsRNAs 1–8 ranged from 34 to 184 nucleotides (nt) and shared 19.1–76.5% identity with each other; the corresponding 3′-UTRs

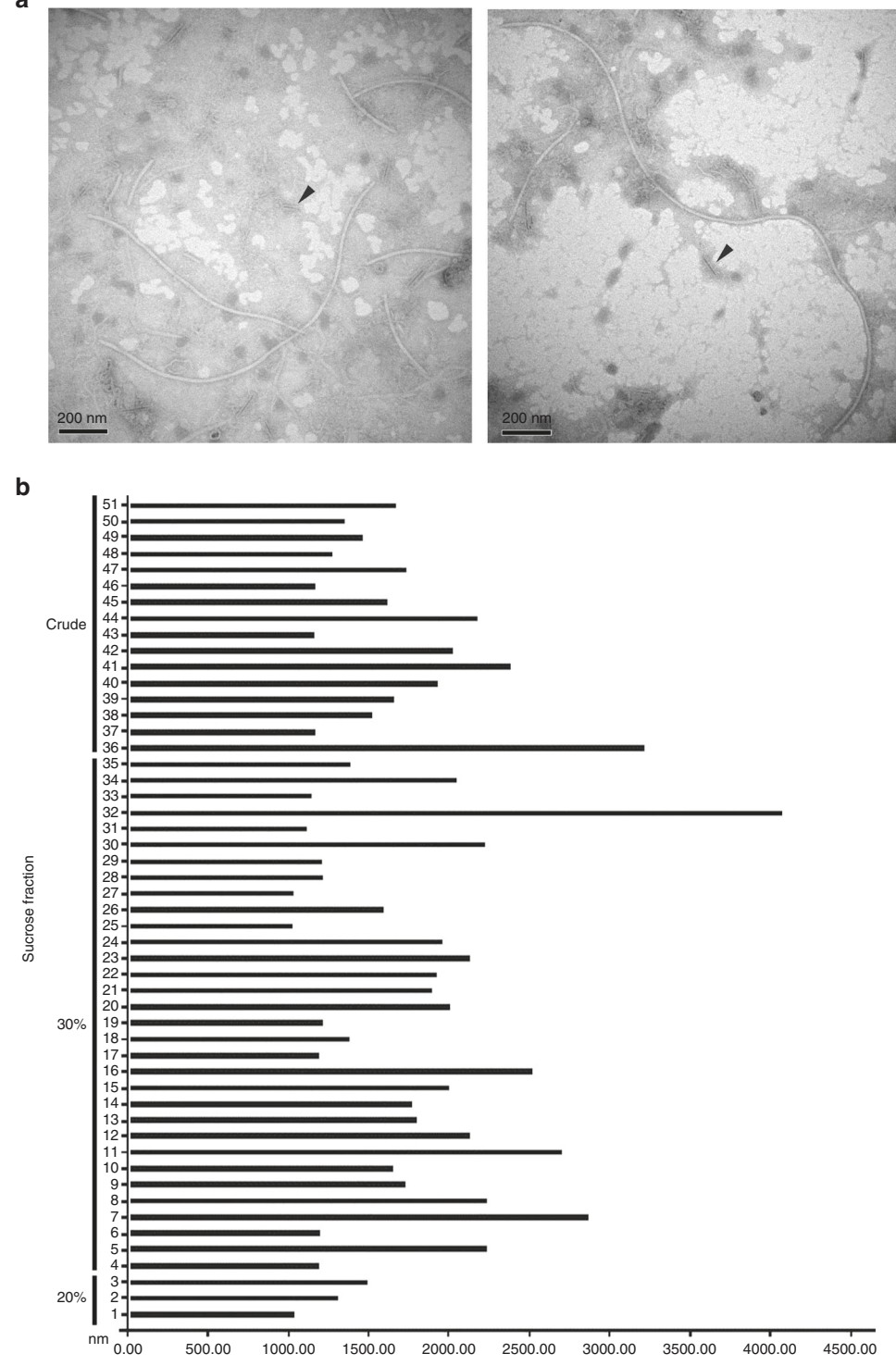

**Fig. 3** Representative transmission electron microscopy (*TEM*) images of virus-like particles extracted from *Colletotrichum camelliae* strain LT-3-1 and a histogram of the sizes of particles longer than 1000.0 nm. **a** Representative virus-like particles extracted from strain LT-3-1 corresponding to the 30% fraction following sucrose gradient centrifugation. The *arrows* indicate empty particles that might have lost dsRNA. *Scale bars*, 200 mm. **b** A histogram of the sizes of particles longer than 1000.0 nm from strain LT-3-1 in crude extracts and in fractions corresponding to 30% and 20% sucrose fractions, respectively, following sucrose gradient centrifugation. The numbers on the *vertical axis* indicate numbers randomly assigned to virus-like particles

ranged from 63 to 339 nt and shared 19.4–45.3% identity (Fig. 1c; Supplementary Table 1). A conserved sequence (G/CGAUAAUAAN$_{(12)}$G/CCUA/UGN$_{(5)}$C), characteristic of dsRNA segmented viruses[1], was observed in all the 5′-termini of the coding strand of dsRNAs 1–8. In contrast, the sequence ACCUUAA was conserved only in dsRNAs 1–5 (Fig. 2a),

and other sequences were also conserved among some of the dsRNAs, e.g., UCCU between dsRNAs 2 and 5, and GAGAAU/ACAGAGU/CG between dsRNAs 7 and 8 (Fig. 2a). No strict sequence conservation was observed in the 3′-termini of the coding strand of the eight dsRNAs. The tetranucleotide CCCC is shared only by dsRNAs 1, 2, 3, and 8, and the sequence

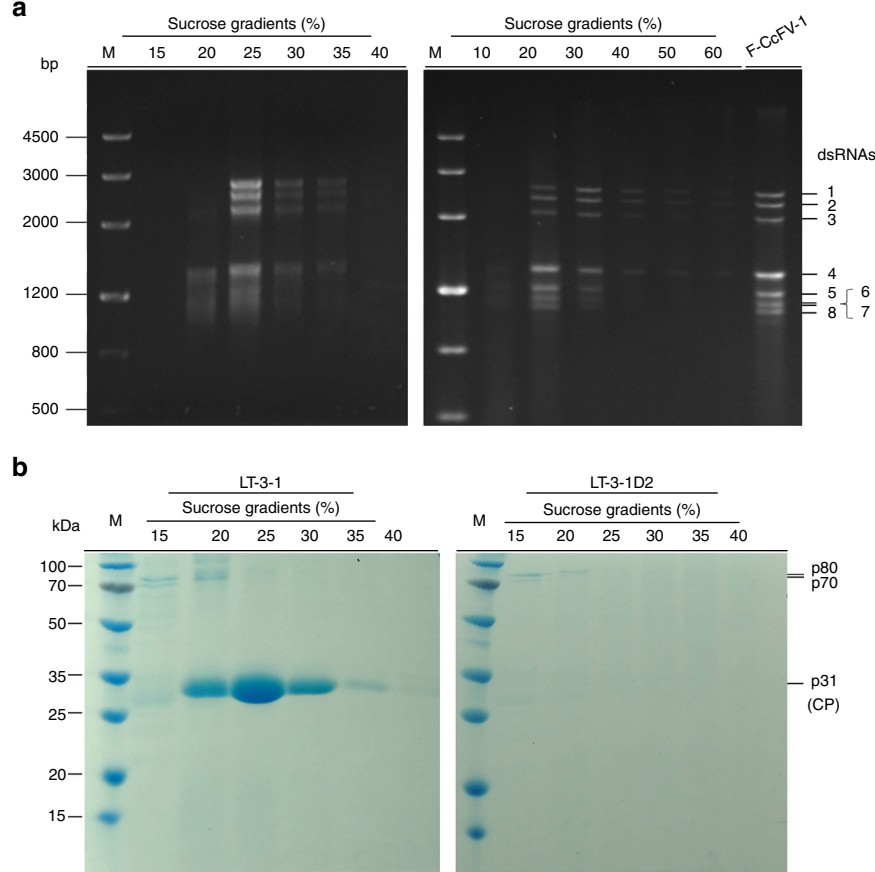

**Fig. 4** Analysis of nucleic acids and proteins associated with virus-like particles. **a** Agarose gel electrophoresis analysis of dsRNAs extracted from purified virus-like particles of *Colletotrichum camelliae* filamentous virus 1 (*CcFV-1*) from 15 to 40% sucrose fractions at 5% increments, from 10 to 60% sucrose fractions at 10% increments, and from mycelia of strain LT-3-1. *F* indicates dsRNAs extracted from fungal mycelia, *M* DNA size marker. **b** SDS-PAGE analysis of proteins extracted from 15 to 40% sucrose gradient fractions (with 5% increments) of strains LT-3-1 and LT-3-1D2. *M* protein molecular weight marker

GGN$_{(6)}$U only by dsRNAs 4, 6, and 7 (Fig. 2b). It is worth noting that dsRNA 5 shares no sequence at the 3′-terminus conserved with the other dsRNAs (Fig. 2b) (confirmed by eight independent 3′-RACE experiments). In addition to the predominant ones, other nucleotides at the termini of each dsRNA were also detected at low frequencies in the RACE experiments, as indicated in Fig. 2a and b together with their frequencies. The sequences conserved in both termini of dsRNAs 1–8 are distinct from those observed in the known related viruses including ATV, CcV-1, AfuTmV-1, BdRV1, BbPmV-1, and BbPmV-2 (e.g., the conserved nucleotides at the 3′-termini for AfuTmV-1, BdRV1, BbPmV-1 and BbPmV-2 are GGGG, UGGGGU, UAGUUUU, UUUU, respectively[9, 12, 15]). Collectively, these data strongly suggest that the dsRNAs 1–8 associated with *C. camelliae* strain LT-3-1 are the genomic components of a novel dsRNA virus, which we propose to name "*Colletotrichum camelliae* filamentous virus 1" (CcFV-1) considering the morphology of their virions (see below).

**Putative proteins encoded by the CcFV-1 genome.** BLASTp searches revealed that the ORFs of dsRNAs 1–4 of CcFV-1 code for proteins that share amino-acid sequence identities of 47–26% (6.0E$^{-171}$–1.0E$^{-27}$, 99–74% coverage) with those encoded by dsRNAs 1–4 of ATV (accession no. KP067914), CcV-1 (KJ787686), AfuTmV-1 (HG975302), and BdRV1 (KP245734) (Supplementary Table 2). The predicted proteins (P1, P2, P3, and

P4) encoded by ORFs 1, 2, 3, and 4 of CcFV-1 share amino-acid sequence identities of 47%, 32%, 26%, and 41% with the RNA-dependent RNA polymerase (RdRp), a hypothetical protein, S-adenosyl methionine-dependent methyltransferase capping enzyme, and a PAS-rich protein (PASrp, containing a high proportion of proline (P), alanine (A), and serine (S) residues) of AfuTmV-1, respectively (Supplementary Table 2). Additionally, protein P2 of CcFV-1 displays a sequence identity of 31% (E-value 1.0E$^{-55}$, coverage 92%) with a hypothetical protein of unknown function from *Alternaria* sp. (ACL80752) (Supplementary Table 2). However, the remaining ORFs of CcFV-1 dsRNAs 5–8 showed no detectable similarity with sequences encoded by viral RNAs (Supplementary Table 2). It is worth noting that CcFV-1 ORF 5 neither shares detectable identity with the ORFs of dsRNA 5 of BdRV1 and CcV-1, nor with those of dsRNA 6 and −7 of BbPmV-2.

Putative protein functions for the CcFV-1 ORFs were also inferred by a homology search using the HHpred server, with the results obtained agreeing with those from BLASTp searches (Supplementary Table 3). CcFV-1 P1 was highly similar to the RdRp of (+)ssRNA viruses belonging to *Caliciviridae* (E-values ≤ 1.3E$^{-47}$) and *Picornaviridae* (E-values ≤ 4.0E$^{-40}$), which infect humans or animals (Supplementary Table 3). Sequence alignment also allowed the identification of three amino-acid motifs (IV, V, and VI) observed previously in AfuTmV-1, BdRV1, BbPmV-1, BbPmV-2, and (+)ssRNA viruses, which are characteristic of supergroup 1 of the viral RdRp family, further supporting that P1

is an RdRp (Supplementary Fig. 1). Interestingly, motif VI also contains the GDN triplet followed by Q rather than G/ADD (which is normally invariant for (+)ssRNA viruses). Moreover, a phylogenetic reconstruction of P1 with RdRps of representative members of dsRNA virus families, including unclassified dsRNA viruses, and of three (+)ssRNA virus families (the closest (+) ssRNA viruses according to BLASTp analysis), clustered CcFV-1 together with AFuTmV-1, BdRV1, BbPmV-1, and BbPmV-2. Such assignment separates these five viruses from other dsRNA viruses, placing them closer to (+)ssRNA viruses belonging to the

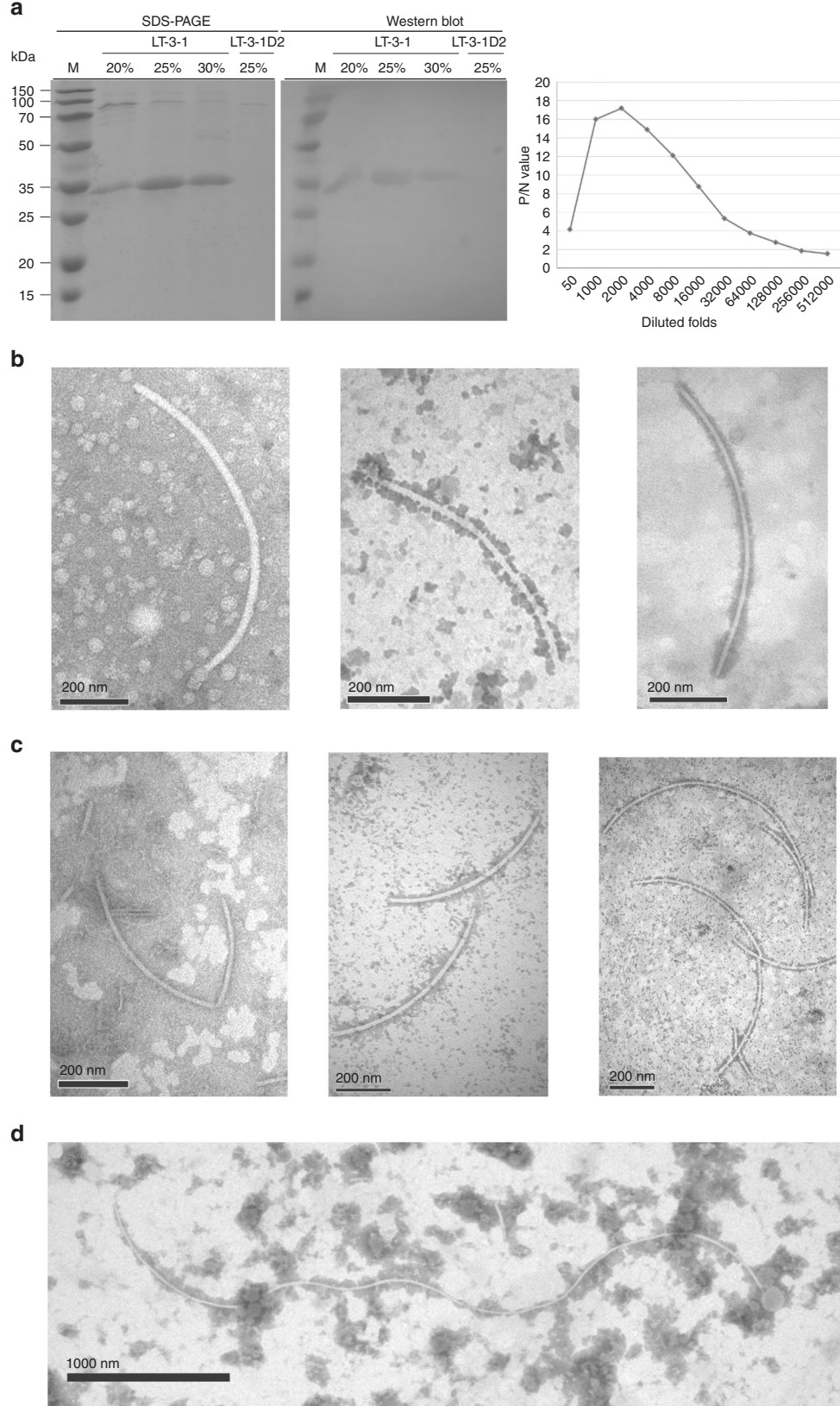

*Caliciviridae*, as predicted by HHpred analysis, in an intermediate position between dsRNA and (+)ssRNA viruses (Supplementary Fig. 2; Supplementary Table 4). These data suggest that CcFV-1 may be an intermediate link between dsRNA and (+)ssRNA viruses, as proposed for AfuTmV-1[9]. On the other hand, CcFV-1 P3 is highly similar to methyltransferase of diverse bacteria (e.g., *Bacillus subtilis*, *Methanococcus maripaludis*, and *Escherichia coli*) (probabilities of 99.8–99.7%, E-values $1.4E^{-17}$–$1.3E^{-15}$), suggesting that P3 is most likely a S-adenosyl methionine-dependent methyltransferase capping enzyme (Supplementary Table 3). No function could be tentatively ascribed to the remaining CcFV-1 putative proteins due to a lack of reliable conserved motifs. Nonetheless, CcFV-1 P4 has a high proportion of P (7.6%), A (13.4%), and S (10.0%) residues, resembling the PASrps encoded by AfuTmV-1 (P 6.8%, A 8.0%, S 9.7%), BdRV1 (P 3.2%, A 9.7%, S 7.9%), BbPmV-1 (P 6.2%, A 13.5%, S 7.6%) and BbPmV-2 (P 11.2%, A 12.2%, S 8.8%)[9, 12, 15].

**Virus-like particles associated with CcFV-1 dsRNAs**. To determine whether the dsRNAs associated with the strain LT-3-1 were encapsidated, mycelial crude preparations were examined by transmission electron microscopy (TEM), which revealed filamentous virus-like particles longer than 127.1 nm (Supplementary Fig. 3a). The presumed viral particles were purified by centrifugation in stepwise sucrose gradients (60–10% with 10% sucrose increments, or 40–15% with 5% sucrose increments), being predominantly found in the gradient zone from 20 to 30% sucrose (Fig. 3). Assuming that the observed short particles were most likely fragments of the longer ones, only those longer than 1000 nm were considered. Measuring 51 filamentous virus-like particles above that limit revealed widths ranging from 11.9 (virus particles number 3, 34, and 35 in Fig. 3b) to 17.7 nm (number 36 and 37) and lengths from 1000.5 (number 25) to 4426.8 nm (number 32) (Supplementary Fig. 3b; Fig. 3b). Some broken particles that appeared empty, most likely resulting from the loss of their dsRNA components, had inner widths ranging from 4.8 to 7.2 nm (Fig. 3a, indicated by arrows). The presumed viral particles were also examined following purification by centrifugation through a CsCl cushion, with the fraction containing the viral dsRNAs and capsid protein (CP) being ultracentrifuged to collect viral particles. TEM examination revealed similar filamentous particles with the longest one having a width of 16.3 nm and a length of 3266.6 nm (Supplementary Fig. 3c). Importantly, removal of the CcFV-1 dsRNA components (dsRNAs 1–8) from the mycelia of strain LT-3-1 when generating a new subisolate (named LT-3-1D2, see below), was accompanied by removal of the filamentous virus-like particles in the crude extract and all sucrose gradient fractions of subisolate LT-3-1D2, thus supporting a close association between them.

**The dsRNAs and proteins composing the virus-like particles**. Agarose gel electrophoresis of the nucleic acids extracted from the 10–60% sucrose gradient fractions (at 10% sucrose increments) showed that the typical pattern of CcFV-1 dsRNAs 1–8 was mostly recovered from the 20 and 30% fractions (Fig. 4a, right

panel). More stringent sucrose gradient centrifugation, with 15–40% sucrose fractions and 5% sucrose increments, confirmed migration of the dsRNAs 1–8 into the 25% fraction; dsRNAs 1–4 were also found in the 20% fraction and dsRNAs 4–8 in the 30 and 35% fractions (Fig. 4a, left panel). These data suggest that dsRNAs 1–8 are separately encapsidated in the virus particles of CcFV-1[16]. No dsRNA bands were recovered from any of the gradient fractions from mycelial extracts of strain LT-3-1D2 performed in parallel.

SDS-PAGE analysis of the proteins from the 40–15% sucrose fractions of strain LT-3-1 revealed the presence of three dominant bands with estimated molecular masses of 80, 70 kDa (p80 and p70, in the 15–20% fractions), and 31 kDa (p31, in the 20–30% fractions). However, while p31 was not observed by SDS-PAGE of the dsRNA-lacking subisolate LT-3-1D2 derived from strain LT-3-1, p80 and p70 were detected in the 15–20% sucrose fractions, thus suggesting that they are host-encoded proteins (Fig. 4b, left panel). To further examine the nature of p80, p70, and p31 (Fig. 4b, left panel, lanes 15–30%), they were eluted from the gel and subjected to peptide mass fingerprinting (PMF). The sequencing data showed no match between the peptide fragments from p80 and p70 (15 and 22, respectively) and the putative proteins encoded by dsRNAs 1–8. Instead, the sequence of these fragments matched proteins from *Colletotrichum* spp., confirming that p80 and p70 are fungal proteins (Supplementary Table 5). In contrast, all peptide fragments (12, 12, and 13 fragments) from p31 preparations separately eluted from 20%, 25%, and 30% sucrose fractions, respectively, matched (scores 692, 723, and 774, respectively) amino-acid regions (60–69%) of the putative P4 protein (Supplementary Table 6). This finding suggested that p31 is the structural protein P4 encoded by CcFV-1 ORF4 (see below).

**P4 forms the capsid of the CcFV-1 virus particles**. P4 was extracted from strain LT-3-1 mycelia, purified from SDS-PAGE, and injected into mice to trigger production of a polyclonal antibody (PAb) against this protein (PAb-P4). The antibody strongly and specifically recognized P4 from 20, 25, and 30% sucrose fractions of strain LT-3-1, while no reactivity was observed with protein extracts from strain LT-3-1D2 (Fig. 5a, left panel). These results indicate that the purified P4 protein used as antigen was indeed uncontaminated with host proteins. Western blotting and indirect enzyme-linked immunosorbent assay (ELISA) analysis revealed a titer of ~128,000-fold dilution for PAb-P4, with an optimum at 2000-fold dilution (Fig. 5a, right panel).

To further confirm that the CcFV-1 dsRNAs were encapsidated, the filamentous virus-like particles prepared by sucrose gradient centrifugation were subjected to immunosorbent electron microscopy (ISEM). The virus-like particles from the 20, 25, and 30% sucrose fractions of strain LT-3-1 were all clearly decorated by PAb-P4 (Fig. 5b, middle and right panels) compared with direct TEM observation (Fig. 5b, left panel). When crude extracts of strain LT-3-1 mycelia were subjected to ISEM analysis the filamentous virus-like particles were also clearly decorated by

**Fig. 5** Western blot analysis and titer quantification of a polyclonal antibody against CcFV-1 P4 and TEM and immunosorbent electron microscopy (*ISEM*) analysis of virus particles. **a** SDS-PAGE (*left panel*) analysis of the proteins from strain LT-3-1 in 20–30% sucrose fractions after sucrose gradient centrifugation and from strain LT-3-1D2 in the 25% sucrose fraction; Western blot analysis of these proteins by SDS-PAGE (*right panel*) using the antibody against CcFV-1 P4 (PAb-P4). Titer quantification of PAb-P4 by indirect ELISA against the P4 protein from fractions after sucrose gradient centrifugation using PAb-P4 diluted from 50- to 512,000-fold. P/N, ratio of the absorbance values of the positive sample (CcFV-1-infected LT-3-1) against the negative sample (virus-free LT-3-1D2) at a wavelength of 450 nm. **b**, **c** Direct TEM and ISEM analysis of virus-like particles derived from 20% and 30% fractions following sucrose gradient centrifugation of strains LT-3-1 (**b**) and LT-3-1T2 (**c**), respectively. The ISEM images indicate that the virus-like particles are decorated by PAb-P4 at a 2000-fold dilution. *Scale bars*, 200 mm. **d** The longest virus-like particles in the crude extract decorated by PAb-P4 at a 8000-fold dilution, as observed by ISEM. *Scale bar*, 1000 mm

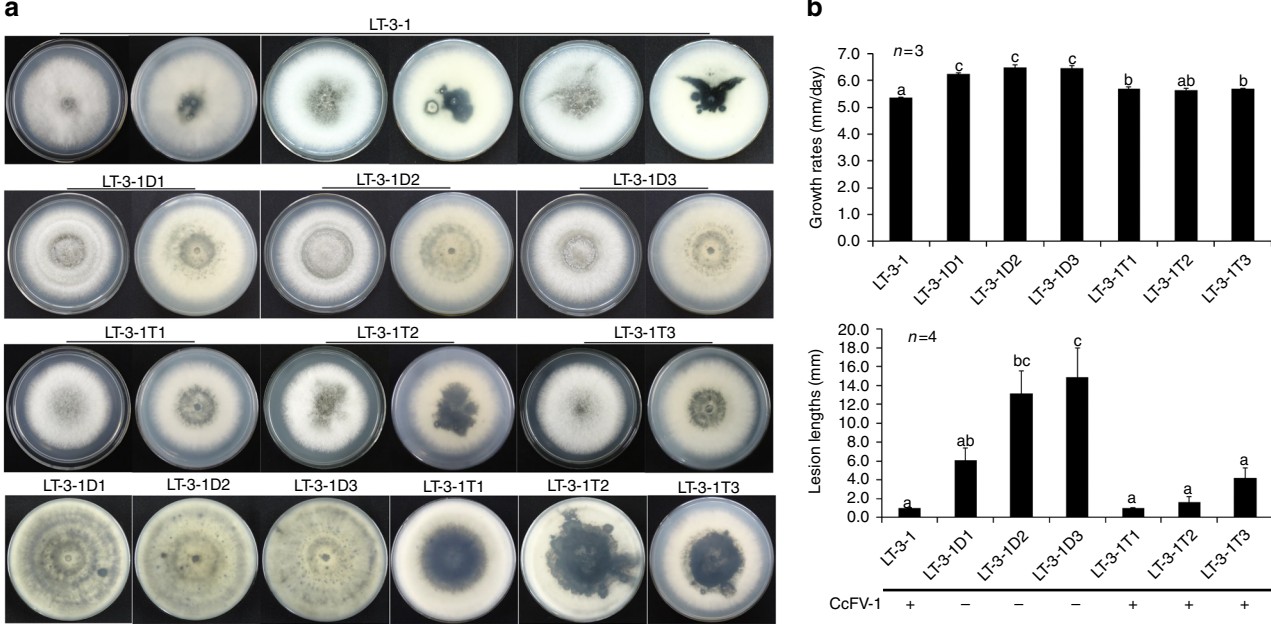

**Fig. 6** Colony morphology, growth rate on PDA, and lesion length of different strains of *Colletotrichum camelliae* on tea leaves. **a** Colony morphology of strain LT-3-1, the LT-3-1 subisolates (LT-3-1D1, -D2, and -D3) with CcFV-1 dsRNAs eliminated, and the subisolates (LT-3-1T1, -T2, and -T3) transfected with dsRNAs 1–8 cultured at 25 °C in the dark at 5 dpi and these subisolates cultured at 18 dpi. **b** A histogram of the growth rates of strain LT-3-1 and the subisolates ($n = 3$) and the lesion lengths induced on tea (*Camellia sinensis* cv. 'Taicha 12') leaves ($n = 4$) inoculated with mycelial plugs of these isolates at 4 dpi under non-wounded conditions. The numbers following the "$n=$" refer to the treatment replicates. The presence of CcFV-1 in these isolates is indicated below the histograms. Symbols "+" and "−" indicate the presence and absence of CcFV-1 based on the results of dsRNA detection obtained by 1.2% agarose gel electrophoresis. Data were analyzed with SPSS Statistics 21.0 (WinWrap Basic; http://www.winwrap.com) by one-way ANOVA, and means were compared using Tukey's test at a significance level of $p = 0.05$. Letters (a, b, and c) over the bars indicate the significant difference at $p = 0.05$. *Bars in each histogram labeled with the same letters are not significantly different ($p > 0.05$). Error bars indicate $\pm$ standard deviation (SD)*

PAb-P4 (Fig. 5d; Supplementary Fig. 4a). However, similar particles were not observed in either crude extracts or in the sucrose gradient fractions from strain LT-3-1D2, supporting that P4 encapsidates the dsRNAs of CcFV-1. Measurements of 28 decorated virus-like particles longer than 1000 nm revealed widths ranging from 12.0 to 17.7 nm (Supplementary Fig. 4b), matching the sizes observed by TEM (Fig. 3b). It is worth noting that among the decorated virus particles in these preparations, the longest had a length of 4661.6 nm (and a width of 17.7 nm, virus particle number 28 in Supplementary Fig. 4b) (Fig. 5d) and the second longest length of 4590.9 nm (and a width of 16.5 nm, number 17) (Supplementary Fig. 4a), both of which were observed in a crude extract. To the best of our knowledge, these are the longest virus-like particles ever reported (Supplementary Table 7).

**CcFV-1 induces phenotypic changes on its fungal host**. Forty-seven subisolates generated from conidial colonies of strain LT-3-1 cultured on potato dextrose agar (PDA) were analyzed by dsRNA electrophoresis and RT-PCR amplification of a 492-bp fragment of CcFV-1 dsRNA 2. CcFV-1 was absent in the subisolates, suggesting no vertical transmission via conidia. The phenotypes of three CcFV-1-free subisolates (LT-3-1D1, LT-3-1D2, and LT-3-1D3) chosen at random, together with their growth rate and virulence, were observed on PDA and measured over a 5-day period. The three CcFV-1-free subisolates exhibited morphologies distinct from the parental strain LT-3-1, including presence of hyphal rings, reduction in black pigments, and normalization of pigment distribution at the bottom of the PDA plate (Fig. 6a). Moreover, the three CcFV-1-free subisolates displayed enhanced growth rates (6.3–6.5 vs. 5.4 mm/day) and

virulence (6.1–14.9 vs. 1.0 mm lesion lengths) compared with strain LT-3-1 (Fig. 6b).

Ultra-thin hyphal sections of strains LT-3-1 and LT-3-1D2 were examined by TEM. Compared with strain LT-3-1D2, the cells of hyphal compartments of strain LT-3-1 showed abnormalities, including formation of large membranous vesicles that might correspond to viroplasms proposed for virus replication and assembly[17], and reductions in Woronin bodies and vacuoles (Supplementary Figs. 5a and b). Virus-like particles with a width similar to those of the CcFV-1 were observed in the vesicles (Supplementary Figs. 5c and d). In addition, ill-defined structures were also observed in the cytoplasm of the hyphal cells of strain LT-3-1 (Supplementary Figs. 5e and f).

**CcFV-1 dsRNAs are infectious and produce virus particles**. To further corroborate that the CcFV-1 dsRNAs were encapsidated in filamentous virus particles, naked dsRNA components were transfected into *C. camelliae* cells to assess whether they could induce synthesis of new filamentous virus-like particles in vivo. Protoplasts prepared from the CcFV-1-free strain LT-3-1D2 were transfected with S1- and DNase I-treated CcFV-1 dsRNAs (1–8) purified from mycelia. A total of 129 subisolates regenerated from *C. camelliae*-transfected protoplasts were randomly subjected to dsRNA electrophoretic analysis, resulting in the presence of all eight transfected dsRNA bands in the mycelial extracts of only three subisolates (named LT-3-1T1, LT-3-1T2, LT-3-1T3), supporting that they were infected by CcFV-1 dsRNAs. When subisolate LT-3-1T2 was subjected to TEM and ISEM analyses, the filamentous virus-like particles were visible in fractions (20–30%) after sucrose gradient centrifugation of mycelial extracts (Fig. 5c, left panel), and were decorated by PAb-P4 (Fig. 5c, middle and right panels). Biological analysis showed

different morphologies, compared with strain LT-3-1D2, for the CcFV-1-transfected subisolates (LT-3-1T1, LT-3-1T2, LT-3-1T3), including disappearance of hyphal rings and increase of black pigments, resembling the colony morphology of strain LT-3-1 (Fig. 6a). The abnormal distribution of black pigments and their high concentration were completely recovered in strain LT-3-1T2 in comparison with strain LT-3-1 (Fig. 6a, middle panel). For the three CcFV-1-infected subisolates, prolonged culture for more than 15 days resulted in the spreading of the black pigments over the bottom of the PDA plate, similarly to strain LT-3-1 but in contrast with all CcFV-1-free isolates including LT-3-1D1 to -D3 (Fig. 6a). The CcFV-1-infected subisolates also exhibited decreased growth rates (5.7 vs. 6.5 mm per day) and virulence (1.0–4.3 vs. 13.1 mm lesion lengths) compared with strain LT-3-1D2, but analogous to strain LT-3-1 (Fig. 6b).

To check whether the CcFV-1 dsRNAs could be transfected into another strain (with a genetic background different from LT-3-1) and promote accumulation of the filamentous particles in *C. camelliae* cells, protoplasts of strain DP-3-1 were also transfected with purified CcFV-1 dsRNAs 1–8. Ten of 94 subisolates gained the tranfected dsRNAs, and one positive subisolate (DP-3-1T28) together with strain DP-3-1, were examined by sucrose gradient centrifugation: the filamentous virus particles were observed in 20–30% sucrose fractions from mycelial extracts of subisolate DP-3-1T28, but not of strain DP-3-1. Parallel experiments in which protoplasts of strains DP-3-1 and LT-3-1D2 were transfected with purified particles, instead of with dsRNAs, failed to provide any positive result.

## Discussion
In this study we have identified a dsRNA virus, CcFV-1, that forms filamentous particles. Several experiments were conducted to confirm that the dsRNAs identified in the fungus *C. camelliae* are indeed encapsided in elongated and flexuous particles. (i) TEM of mycelial preparations from strains infected by CcFV-1 revealed filamentous particles in the sucrose gradient fractions in which the structural proteins and the genomic dsRNAs migrated, while the virus particles and structural proteins were absent in strains in which the dsRNAs were removed. (ii) PAb-P4, a polyclonal antibody against the purified structural protein P4 encoded by CcFV-1 ORF4, specifically recognized a structural protein of ~31 kDa in specific sucrose gradient fractions, whereas no reactivity with host proteins was found; moreover, PAb-P4 decorated the filamentous particles in crude extracts and sucrose gradient fractions of CcFV-1-infected strains or subisolates. (iii) Transfection of naked CcFV-1 dsRNAs into protoplasts of strains LT-3-1D2 and DP-3-1 promoted the accumulation of filamentous virus particles in *C. camelliae* cells, further supporting that the genomic dsRNAs are encapsidated in these particles. The lack of infectivity of the filamentous particles possibly resulted from technical difficulties in their preparation or transfection. Taken together, these data provide strong evidence that the mycoviral dsRNAs are encapsidated by P4 in filamentous virions. Typical dsRNA viruses encapsidate a single-stranded pregenome together with the RdRp, with the second strand being synthesized within the virion. Here, SDS-PAGE analysis failed to reveal the RdRp in association with the CP. We propose that the viral dsRNAs could be encapsidated following synthesis of the second strand in the host cell. Moreover, like viral (+) ssRNAs, the CcFV-1 genomic dsRNAs are infectious, suggesting that CcFV-1 may represent an intermediate stage in the evolution of a (+)ssRNA virus into a dsRNA virus.

Members of seven families of dsRNA viruses have been analyzed by cryo-electron microscopy or crystallography. With the exception of birnaviruses and mycoreoviruses, all other dsRNA viruses (including those that infect prokaryotes and complex eukaryotes) share 120-subunit T = 1 icosahedral capsids ranging from 25 to 50 nm in diameter[18]. Therefore, the present study is, to the best of our knowledge, the first report that dsRNA viruses may assemble into elongated and flexuous particles (Supplementary Table 7), which in the case of CcFV-1 particles are the longest found for any known filamentous virus (Supplementary Table 7). It is surprising that such small dsRNA molecules (990–2444 bp) are encapsidated into large particles of 4661.6 nm, suggesting distinct packaging or encapsidation strategies. Furthermore, the related viruses AfuTmV-1 and BbPmV-1 also show unusual structures, since the infectious entity appears associated with PASrp in a colloidal form[9, 12]. BdRV1, however, might be capsidless or form bacilliform virus-like particles[15]. Therefore, there is a major discrepancy in virion morphology among these related dsRNA viruses, illustrating the very diverse nature of virion architecture with this virus group. Further studies are needed to determine whether a novel genome packaging mechanism or protein subunit arrangement is involved.

The putative P1 protein of CcFV-1 is predicted to be an RdRp that additionally contains the GDNQ motif characteristic of the L genes of rhabdoviruses and paramyxoviruses within the order *Mononegavirales*[19] (Supplementary Fig. 1). A similar feature was previously reported in the dsRNA viruses AfuTmV-1, BdRV1, BbPmV-1, and BbPmV-2[9, 12, 15], suggesting that, like the GDD motif in (+)ss RNA viruses, the RdRp GDNQ motif of CcFV-1 could have important functions in metal ion coordination and nucleotide substrate binding during dsRNA replication. The putative protein P3 is predicted to be a S-adenosyl methionine-dependent methyltransferase capping enzyme (Supplementary Table 3), as suggested for AfuTmV-1 and BbPmV-1, and it might be involved in a cap-snatching mechanism[9, 20]. The combination of a capping enzyme with a picorna-like RdRp is unusual in a viral genome. According to SDS-PAGE, PMF, and ISEM analyses, P4 protein (p31) is the CcFV-1 CP encapsidating the viral genome into filamentous particles. The functions of the other CcFV-1 putative proteins remain obscure due to a lack of sequence similarity with known proteins.

Previous studies have shown that RdRps from the dsRNA virus groups display features similar to those of different (+)RNA viruses, leading to the conclusion that dsRNA viruses may have evolved from distinct supergroups of (+)RNA viruses[9, 21]. In this context, the RdRp of CcFV-1 appears in an intermediate position between those of (+)ssRNA viruses (calicivirus, picornavirus, and astrovirus) and dsRNA viruses, being closer to that of calicivirus (Supplementary Fig. 2). Therefore, CcFV-1 appears as a candidate linking dsRNA and (+)ssRNA viruses and it might have originated from a (+)ssRNA viral ancestor belonging to clades three or six of the picorna-like superfamily[5]. However, CcFV-1 CP has no detectable sequence similarity with proteins from known filamentous (+)ssRNA viruses that infect plants, including members of the *Flexiviridae*, *Potyviridae*, and *Closteroviridae*, suggesting that it should not have originated from the same single ancestral protein[7]. Interestingly, the CcFV-1 phylogenetically related dsRNA viruses AfuTmV-1 and BbPmV-1, which were isolated from human and insect pathogenic fungi, respectively, lack a capsid[9, 12]. Therefore, CcFV-1 features support a relationship among mycoviruses from fungi infecting humans, insects, and plants, as well as between encapsidated and capsidless dsRNA viruses. It is worth noting that P4 is also a PASrp that resembles those encoded by the dsRNA4 of AfuTmV-1 and BbPmV-1. However, the PASrp of AfuTmV-1 and BbPmV-1 apparently coat but do not encapsidate the viral genome as visualized by atomic force microscopy[9, 12].

Therefore, CcFV-1 appears to be a specimen linking filamentous and capsidless dsRNA viruses, with the PASrp supporting the evolution trace between capsidless and capsided viruses. Furthermore, AfuTmV-1, BdRV1, and BbPmV-1 have four or five genomic dsRNA components[9, 15], and BbPmV-2 and -3 seven[12], whereas CcFV-1 has eight, of which dsRNAs 5–8 contain ORFs with no detectable identity with those of other viruses based on available sequences, including the fifth dsRNA of BdRV1 and CcV-1 (Supplementary Table 7), thus supporting the idea that these viruses have a dynamic genomic organization in terms of segment number and sequence[12]. Collectively, CcFV-1 exhibits a complex evolutionary lineage, suggesting that it is the result of a unique evolutionary process distinct from those of known dsRNA viruses.

CcFV-1 can impair host cellular homeostasis, confer hypovirulence, and impact host morphology, including pigment production and distribution, growth rate and hyphal ring generation. However, the virus-infected fungal strains maintained high growth activities, unlike the situation found with many other mycoviruses where hypovirulence severely impairs activity[10]. These features would benefit virus survival and spread in the field in potential applications of CcFV-1 as a fungicide.

In summary, to the best of our knowledge, this is the first report of a dsRNA virus that is encapsidated within filamentous particles, which are the longest among known viral particles. Our finding is a significant addition to the evolutionary genomic diversity and particle architecture of the expanding virosphere. CcFV-1, with its unique molecular features, should contribute to our understanding of the genomic function, ecology, and evolution of dsRNA viruses.

## Methods

**Fungal isolates**. Strains LT-3-1 and DP-3-1 of *C. camelliae* Massee were isolated from tea leaves (*C. sinensis* (L.) O. Kuntze) collected in Yichang, Hubei Province, China, and identified based on molecular and morphological analyses. Both strains were purified using the hyphal-tipping technique[22].

**Extraction of the dsRNAs and enzymatic treatments**. For dsRNA extraction, mycelial plugs were inoculated in cellophane membranes on PDA (20% diced potatoes, 2% glucose, and 1.5% agar) plates at 25 °C in the dark for 4–5 days. The mycelia were collected, ground to a fine powder in liquid nitrogen, and subjected to dsRNA extraction and protein elimination using a patented method developed in our lab[23]. Briefly, the frozen powder was mixed with five volume of SDS buffer (2% SDS, 4% PVP-40, 0.5 M NaCl, 100 mM Tris-HCl (pH 8.0), 20 mM EDTA (pH 8.0)), and centrifuged at 12,000 × g for 5 min to collect the supernatant, which was mixed with an equal volume of binding buffer (50% guanidine thiocyanate, 1.5 M KCl, 0.5 M NH₄Cl, 0.3 M KAcO (pH 6.0)) and centrifuged at 12,000 × g for 5 min to remove host proteins. The resulting supernatant was mixed with 0.5 volume of ethanol, and loaded into a silica spin column (Sangon Biotech (Shanghai) Co., Ltd, China) to absorb the dsRNA. Following two washes with the binding buffer containing 37% ethanol, the dsRNAs were eluted with RNase-free water.

The dsRNA preparations were subsequently treated to remove the remnant ribosomal RNAs and DNAs. Briefly, aliquots of 200 ng of total dsRNA were digested with 2 U DNase I (New England Biolabs) at 37 °C for 1 h, treated with phenol/chloroform/isoamyl alcohol (25:24:1) (phenol saturated with water, pH 5.2), precipitated with ethanol, dissolved in diethylpyrocarbonate (DEPC)-treated water, and treated with 10 U S1 nuclease (Thermo Scientific) at 37 °C for 1 h. The purified RNAs were analyzed by 1.2% agarose gel electrophoresis and visualized by staining with ethidium bromide. Each dsRNA was separately excised and purified using a gel extraction kit (Qiagen, USA), dissolved in DEPC-treated water and stored at −70 °C until use.

The dsRNA preparation together with BdCV 1 dsRNAs (extracted from mycelia infected by BdCV 1)[23] and in vitro transcripts of dimerized cDNAs of CEVd (CEVd.188)[4] were treated with S1 nuclease (Thermo Scientific) as indicated above, 0.4 U RNase III (New England Biolabs) in cleavage buffer (30 mM Tris (pH 8.0), 160 mM NaCl, 0.1 mM EDTA, 0.1 mM DTT, 10 mM MgCl₂), and 200 ng/ml RNase A (Thermo Scientific) in 2× SSC (300 mM NaCl, 30 mM sodium citrate, pH 7.0) or 0.1× SSC as previously described[2, 24].

**Cloning and sequencing**. The cDNA sequences of genomic dsRNAs were determined as previously described[14]. The 5′- and 3′-terminal sequences of the

dsRNAs were obtained from dsRNAs extracted and individually separated from purified virus particles (see below), as previously described[13]. At least three independent clones of each fragment were sequenced in both directions. Sequencing was performed by Sangon Biotech (Shanghai) Co., Ltd, China, and every nucleotide was determined in at least three independent overlapping clones in both orientations. The full-length of the shorter dsRNAs (dsRNAs 7 and 8) were also determined as previously described[3] with minor modifications. Briefly, the 3′ terminus of each dsRNA strand was ligated with the closed adaptor primer RACE-OLIGO with a phosphorylated (p) 5′ end and an NH2 3′ end (5′-p-GCATTGCATCATGATCGATCGAATTCTTTAGTGAGGGTTAATT GCC-(NH2)-3′), reverse transcribed with M-MLV reverse transcriptase (Promega Corporation) using the oligo REV (5′-GGCAATTAACCCTCACTAAAG-3′, complementary to the adaptor at positions 46–26), amplified with PCR using primer O5RACE-2 (5′-TCACTAAAGAATTCGATCGATC-3′, complementary to the adaptor at positions 34–17), followed by nested PCR amplification using O5RACE-3 (5′-CGATCGATCATGATGCAATGC-3′, complementary to the adaptor at positions 17–1), and cloned as described above.

Sequence similarity searches were performed using National Center for Biotechnology Information (NCBI) databases with the BLAST program. Multiple alignments of nucleic and amino-acid sequences were conducted using MAFFT version 6.85, as implemented at http://www.ebi.ac.uk/Tools/msa/mafft/ with default settings except for refinement with 10 iterations. Identity analyses were conducted using the MegAlign program (version 5.00) with the Clustal W method (DNASTAR Inc.). The phylogenetic tree for RdRp sequences was constructed as previously described[23]. ORFs were deduced using ORFfinder (https://www.ncbi.nlm.nih.gov/orffinder/). Functions of the deduced proteins were predicted with hidden Markov models implemented in the HHpred (Homology Detection & Structure Prediction by HMM-HMM comparison) package (http://toolkit.tuebingen.mpg.de/hhpred)[25].

**Purification of virus particles from mycelia**. A mycelial plug was grown at 25 °C in the dark for 8 days in sterilized cellophane membranes placed on PDA. After harvest, 30 g mycelia was ground to a fine powder in liquid nitrogen, and the resulting powder was subjected to extraction. Briefly, the powder was mixed with 100 mM phosphate buffer (PB; 8.0 mM Na₂HPO₄, 2.0 mM NaH₂PO₄, pH 7.0) and centrifuged at 12,096 × g at 4 °C for 30 min to remove cellular debris. The supernatant was further ultracentrifuged (Optima LE-80K; Beckman Coulter, Inc.) at 110,000 × g at 4 °C for 1 h to collect the sediment, which was resuspended in 100 mM PB buffer. The crude extract was further subjected to virus purification following sucrose gradient centrifugation as previously described[23], or by centrifugation at 55,000 × g at 4 °C for 1.5 h through a cushion of CsCl (1.45 g/cm³), as previously described[9]. The crude or purified virus particle preparations were negatively stained with 1% uranyl acetate on carbon-coated 400-mesh copper grids and examined by TEM (H-7000FA; Hitachi) before storage at −70 °C. The widths and lengths of the particles were measured using ImageJ 1.43[26], and the width of each particle was determined based on at least five measurements along the particles selected at random.

**Analysis of the dsRNAs and proteins from the viral particles**. Purified virus-like particles were subjected to viral dsRNA extraction and analysis as previously described[23]. Briefly, 200 μl sucrose suspension was collected from each fraction after sucrose gradient centrifugation and treated with phenol/chloroform/isoamyl alcohol (25:24:1) (pH 5.2) to remove viral proteins. The nucleic acids were precipitated with ethanol, dissolved in DEPC-treated water and analyzed by agarose gel electrophoresis.

Proteins extracted from each sucrose fraction were analyzed by 12% SDS-PAGE with 25 mM Tris-glycine and 0.1% SDS. Following electrophoresis, the gels were stained with Coomassie brilliant blue R-250 (Bio-Safe CBB; Bio-Rad, USA). The protein bands on the gel were then individually excised and subjected to PMF analysis by Sangon Biotech (Shanghai) Co., Ltd, China, as previously described[27].

**Ethics statement**. This work was approved by the Research Ethics Committee, Huazhong Agricultural University, Hubei, China (HZAUMO-2016-040), and carried out in accordance with the recommendations in the Guide for the Care and Use of Laboratory Animals from this committee.

**Polyclonal antibody production and ISEM examination**. The P4 protein was extracted from the CcFV-1-infected strain LT-3-1 and purified using sucrose gradient centrifugation, as previously described[23]. After separation by SDS-PAGE, as described above, the gel bands containing P4 were excised for antibody preparation. A total of 300 μg of P4 was injected four times into two 3-week-old female BALB/c mice (purchased from the Laboratory Animal Centre, Huazhong Agriculture University, Hubei Province) to generate polyclonal antibodies (PAb-P4), as previously described[28].

Protein preparations from CcFV-1-infected and -free isolates were extracted from sucrose fractions after sucrose gradient centrifugation, as previously described[23]; the proteins were then subjected to indirect ELISA, as previously described, with PAb-P4 dilutions ranging from 1:50 to 1:512,000 to estimate the

best titer[15]. Western blotting and ISEM analysis were also performed as previously described[29] with PAb-P4 at dilutions ranging from 200- to 8000-folds.

**Elimination of CcFV-1 dsRNAs.** Mycelial plugs of *C. camelliae* strain LT-3-1 were grown on PDA at 25 °C under a 24-h photoperiod for more than 1 month until sporulation. At the end of the culture, the conidia were collected and cultured on PDA plates at 25 °C in darkness for 24 h. The small colonies that developed were separately transferred to new PDA plates for dsRNA extraction. The extracted dsRNA were analyzed by 1.2% agarose gel electrophoresis, visualized by staining with ethidium bromide, and then further identified by RT-PCR.

RT-PCR amplification was performed using a specific primer pair derived from the dsRNA 2 sequence (RNA2-1-F: 5′-ACGCCTTTACATATCTGTTGTCA-3′; RNA2-2-R: 5′-CGTGCGTAGTAGCTCAACCCT-3′), generating a 492-bp fragment. An annealing temperature of 57 °C was used with a PCR Thermal Cycler (Model PTC-100, MJ Research, USA).

**Analysis of the biological features of the fungal strains.** Morphology and growth rates were estimated by culturing 4-day-old mycelial plugs on PDA in triplicate at 25 °C in the dark for 3–5 days, as described in our previous report[23]. The virulence of each strain was determined by inoculating mycelial plugs on detached tea leaves of cultivar 'Taicha 12' (*C. sinensis* cv. 'Taicha 12') using four replicates, as previously reported[23]. The lesions that developed were measured and photographed at 4 days post inoculation (dpi).

Ultra-thin sections (50–60 nm in thickness) of mycelial specimens of *C. camelliae* strains LT-3-1 and LT-3-1D2 were cut using a ultramicrotome (EM UC7; Leica), mounted on slotted formvar-coated grids, stained with 5% aqueous lead citrate and 5% uranyl acetate, and observed by TEM (Tecnai G2 20 TWIN; Fei).

**Transfection with CcFV-1 dsRNAs and virus-like particles.** Total dsRNA (1–8) was extracted from mycelia of strain LT-3-1 and analyzed as described above. Aliquots of 2.0 μg of total dsRNA were digested with 20 U DNase I and 100 U S1 nuclease (New England Biolabs), as described above, dissolved in DEPC-treated water, and stored at −70 °C until use.

Protoplasts were prepared from actively growing mycelia of *C. camelliae* strain LT-3-1D2, a subisolate regenerated from a conidium of isolate LT-3-1, or strain DP-3-1 as previously described[30]. The protoplasts were filtered through a Millipore filter, counted under a microscope (×100) using a hemocytometer, and used for dsRNA transfection using PEG 6000 as previously described[31]. A total of $2.0×10^6$ protoplasts were transfected with ~5.0–7.0 μg of enzyme-treated dsRNAs (containing dsRNAs 1–8) or 70.0–80.0 μg virus-like particles. Following transfection, the protoplast suspensions were diluted with sterilized water and then spread onto PDA plates for colony formation. The new colonies were separately cultured on new PDA plates for dsRNA extraction.

**Data analysis.** Descriptive statistics were determined, and $\chi^2$-tests, one-way analysis of variance, and Tukey post hoc tests were performed using SPSS Statistics 17.0 (WinWrap Basic; http://www.winwrap.com). $p < 0.05$ was considered to indicate statistical significance.

**Data availability.** Sequence data supporting the findings of this study have been deposited in GenBank under accession numbers KX778766-KX778773 for dsRNAs 1–8 of CcFV-1, respectively. The remaining data are available within the article and its Supplementary Information files and from the corresponding author upon request.

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

## Acknowledgements

This work was supported by grants from the Fundamental Research Funds for the Central Universities (no. 2662017PY026) to W.X. We thank Prof. Ricardo Flores, Instituto de Biología Molecular y Celular de Plantas, (IBMCP, UPV-CSIC), Valencia, Spain, for remarkably improving the manuscript. We also thank Prof. Said A. Ghabrial, Department of Plant Pathology, University of Kentucky, Lexington, KY 40546-0312, USA, for kindly reviewing the manuscript. We are grateful to Bichao Xu of the Core

Facility and Technical Support, Wuhan Institute of Virology for her technical support with regard to transmission electron microscopy.

## Author contributions

W.X.: Designed the investigation and wrote the manuscript. H.J.: Performed most of the experiments. K.D.: Conducted the enzyme treatment and part of the transfection and virulence analyses. L.Z.: Involved in transfection and virulence analyses. J.D.: Wrote part of the abstract and discussion. N.H. and G.W.: Involved in the design of the investigation.

## Additional information

**Competing interests:** The authors declare no competing financial interests.

