## [Peer Review File · Nature Communications]

Reviewers' comments:

Reviewer #1 (Remarks to the Author):

This paper describes a very novel virus from a filamentous fungus, that is reported to have a segmented double stranded RNA genome, and be encapsidated in flexuous rods. There are some very surprising features of this virus: most multicomponent flexuous rod viruses are packaged separately; no dsRNA virus has been found packaged in a flexuous rod, which fits with what is known about their biology; and the dsRNA is infectious. All of these points are difficult to explain in the context of what is known about virus particles, packaging, and especially the life cycle of dsRNA viruses.

The writing needs a lot of work. The authors need to either pay for editorial services or get some help from a native English speaker. It is not acceptable as it is written.

I am skeptical that this is really a dsRNA virus for several reasons: one of the hallmarks for placing a virus into a particular class is the RdRp, and this virus appears to have a ssRNA-dependent RNA polymerase. For example both the Hypoviruses and the Endornaviruses are now usually considered ssRNA viruses due to their RdRp, the replicative intermediates are the dsRNA elements that are isolated due to the lack of real packaging. In addition, the authors do not describe their method for dsRNA extraction (neither here nor in their previous paper), so it is not possible to assess this. Their nuclease digestions do not contain any controls of ssRNA, and usually dsRNA is demonstrated by resistance to RNase A. S1 is really a DNase, although it has some activity on ss RNA. Given the very unusual (and biologically very surprising and inexplicable) finding of a filamentous dsRNA virus, these need to be very thoroughly investigated, and more should be done to unequivocally prove the dsRNA nature of the genome. Some additional specific comments are below.

l. 52 to 53, this is not clear (I don't know what it means). The references are only for prokaryotic viruses.

l. 68-70, there are some inaccuracies here, for example some partitiviruses have more than 2 segments, and chrysovirus often have 3 segments.

l. 72 to 75, viruses characterized as unencapsidated dsRNA viruses are almost certainly the replicative forms of ssRNA viruses.

l, 89-90, you cannot determine sizes this precisely from a gel!

l. 96-97, how do you know that you actually have the complete RNA?

l. 107-8, 31% identity is not "considerably high" (sic), especially when a higher level of similarity is considered "low" a few sentences later!!

l. 139-146, (and figure 2C) computer generated secondary structures have little relevance

without corroborating data, I recommend deleting these. They are not put into any context; since these are reported as dsRNA viruses, what is the relevance of the ss structure?

The discussion does not propose any mechanisms for some of the inexplicable points above. All known dsRNA viruses encapsidate a ss pregenome into a virion that contains the RdRp, and the second strand is synthesized in the virion. How could this happen for this virus? Purified viral RNA from dsRNA or even (-) ssRNA viruses has never been found to be infectious, because these viruses require an RdRp to establish infection, they cannot make it directly from their genomic RNA as ssRNA viruses can. How can they explain this? The discussion needs to be expanded to include these points.

Reviewer #2 (Remarks to the Author):

This reviewer assessed a similar version of the manuscript, submitted earlier to another journal, basically with the same contents by the same authors. The authors incorporated many of my previous suggestions into the current version.

In the paper no NCOMMS-17-01167, the authors report a characterization of a novel fungal virus termed *Colletotrichum camelliae* filamentous virus 1 (CcFV-1). The virus was isolated from a phytopathogenic fungus (strain LT-3-1) that had a 8-segmented dsRNA genome. Findings include: 1) detection of CcFV1 from a fungus unreported as a virus host, 2) hypovirulence conferred by CcFV1 to the fungal host, 3) CcFV1 as an evolutionary intermediate between (+)RNA and dsRNA viruses, 4) the GDNQ motif of CcFV1 RdRp typically possessed by mononegaviruses, 5) extremely long, filamentous particles accommodating CcFV1 dsRNA genomic segments with the dsRNA4-encoded protein, and 6) infectivity of viral dsRNA fractions treated by nuclease S1 and DNase I.

The fact that CcFV1 is closely related to two previously reported fungal viruses: *Aspergillus fumigatus* tetramycovirus 1 (AfuTmV-1) (Kanhayuwa et al., PNAS 112, 2015) and *Botryosphaeria dothidea* virus 1 (BdRV1) (Zhai et al., Virology 493, 2016) lowers the scientific impact of the current paper. Also, recently multiple related viruses were reported by Kotta-Loizou and Coutts (PLoS Pathogen, 2017). Interestingly, they have different genome segment numbers ranging from 5-7 unlike AfuTmV-1 and BdRV1 that have 4 and 5 genomic segments, respectively. Thus the genome segment number 8 of CcFV1 is not as surprising as the authors underline.

As a whole, I feel that CcFV-1's filamentous particle nature and infectivity as dsRNA are of sufficient scientific merit for Nature Communication. However, the first point should be substantiated in a solid way. There follow a few major and minor suggestions:

Major points:

1. As evident from the title of the paper, the most appealing point is the filamentous structure of CcFV1. The authors showed it by ISEM using antibodies to gel-purified 31-kDa protein and confirmed the presence of similar structures in thin sections of infected cells.

Given the potential high impact, this notion should be reinforced by another approach. For example, immune-gold labeling of thin sections or immune-trapping of filamentous particles from infected mycelia homogenates, not from purified fractions, should be performed. A related issue is particle length distribution. The authors have provided a sort of histogram (Fig. 3), but the number of tested particles is limited. They should increase them.

2. No comparison in virion structure is made among the closely related and characterized viruses AfuTmV-1, BdRV1 and CcFV-1. Zhai et al. (Virology 493, 2016) reported "bacilliform virus-like particles about 30–80 nm in length and 13.4 nm in diameter" as presumable BdRV1 virions, although they failed to prove it. Kanhayuwa et al. (PNAS 112, 2015) showed the unusual AfuTmV-1 structure as an infectious entity associated with P(proline)-A(alanine)-S(serine)-rich protein in a colloidal form. Therefore, there is a huge discrepancy in virion morphology among these closely related "tetraviruses." The authors should mention this in an appropriate place. Also I would suggest the authors to test the purification methods of Zai et al. and Kanhayuwa et al. for CcFV1 and state their results.

3. Kotta-Loizou and Coutts recently reported viruses with 4-7 dsRNA genomic segments closely related to AfuTmV1 (PLoS Pathogens). A thorough comparison should be made and briefly mentioned in an appropriate place.

4. Perhaps I missed it, but did the authors test "filamentous" particles for infectivity? The authors should touch whatever its outcome may be.

4. Condense pages 6-9.

Response to the reviewers

Attached please find our response, highlighted with yellow background, to the reviewers comments item by item.

Reviewer #1 (Remarks to the Author):

This paper describes a very novel virus from a filamentous fungus, that is reported to have a segmented double stranded RNA genome, and be encapsidated in flexuous rods. There are some very surprising features of this virus: most multicomponent flexuous rod viruses are packaged separately; no dsRNA virus has been found packaged in a flexuous rod, which fits with what is known about their biology; and the dsRNA is infectious. All of these points are difficult to explain in the context of what is known about virus particles, packaging, and especially the life cycle of dsRNA viruses.

The writing needs a lot of work. The authors need to either pay for editorial services or get some help from a native English speaker. It is not acceptable as it is written.

I am skeptical that this is really a dsRNA virus for several reasons: one of the hallmarks for placing a virus into a particular class is the RdRp, and this virus appears to have a ssRNA-dependent RNA polymerase. For example both the Hypoviruses and the Endornaviruses are now usually considered ssRNA viruses due to their RdRp, the replicative intermediates are the dsRNA elements that are isolated due to the lack of real packaging. In addition, the authors do not describe their method for dsRNA extraction (neither here nor in their previous paper), so it is not possible to assess this. Their nuclease digestions do not contain any controls of ssRNA, and usually dsRNA is demonstrated by resistance to RNase A. S1 is really a DNase, although it has some activity on ss RNA. Given the very unusual (and biologically very surprising and inexplicable) finding of a filamentous dsRNA virus, these need to be very thoroughly investigated, and more should be done to unequivocally prove the dsRNA nature of the genome.

RESPONSE. We provide substantial evidence to prove CcFV-1 is a dsRNA virus: i) after digestion with RNase S1, all the single-stranded ribosomal RNAs and mRNAs were removed from the gel, but the CcFV-1 dsRNAs remained intact (Fig. 1a), ii) the phylogenetically related viruses are classified as dsRNA viruses, i.e., *Aspergillus fumigatus* tetramycovirus 1 (AfuTmV-1) (Kanhayuwa et al., PNAS 112, 2015) and *Botryosphaeria dothidea* virus 1 (BdRV1) (Zhai et al.,

Virology 493, 2016), and iii) CcFV-1 dsRNAs are resistant and sensitive to RNase A in a high and low ionic strength, respectively. More specifically, CcFV-1 dsRNAs are resistant against RNase A in 2xSSC and but sensitive to RNase A in 0.2xSSC; we have run parallel controls with a viral dsRNA and a single-stranded RNA. We do not consider necessary to include these data in manuscript. Instead, we provide them as supplementary data for review.

Some additional specific comments are below.

1. 52 to 53, this is not clear (I don't know what it means). The references are only for prokaryotic viruses.

RESPONSE. We have changed the expression as: “The morphotypical peculiarities of viruses have been shaped by the environment and the specific nature of their hosts, as particularly noticeable in archaeal viruses”.

"1. 68-70, there are some inaccuracies here, for example some partitiviruses have more than 2 segments, and chrysovirus often have 3 segments.

RESPONSE. We have checked and improved the information here, including correction of the segment numbers as 2-3 for partitiviruses, and 3-5 for chrysovirus.

1. 72 to 75, viruses characterized as unencapsidated dsRNA viruses are almost certainly the replicative forms of ssRNA viruses.

RESPONSE. We have commented on this point since we have not found such statement in published papers. Instead, excepting *Endornaviridae*, other unencapsidated dsRNA viruses including *Aspergillus fumigatus* tetramycovirus-1 (AfuTmV-1), and *Beauveria bassiana* polmycovirus-1, -2 and -3 (BbPmV-1, -2 and -3) have been identified as dsRNA viruses [Kanhayuwa, L., Kotta-Loizou, I., Özkan, S., Gunning, A. P. & Coutts, R. H. A. A novel mycovirus from *Aspergillus fumigatus* contains four unique dsRNAs as its genome and is infectious as dsRNA. Proc Natl Acad Sci USA 112, 9100-9105 (2015); Kotta-Loizou, I. & Coutts, R. H. A. Studies on the virome of the entomopathogenic fungus *Beauveria bassiana* reveal novel dsRNA elements and mild hypervirulence. PLoS Pathog 13, e1006183 (2017)].

l, 89-90, you cannot determine sizes this precisely from a gel!

RESPONSE. We have changed the text as “approximately 2500 and 900 bp”.

l. 96-97, how do you know that you actually have the complete RNA?

RESPONSE. Here we had repeatedly analyzed the dsRNA segments by PAGE and carefully checked the contigs that have been clustered in known dsRNA sequences. From these data we have concluded that we have reconstructed all the dsRNA segments.

l. 107-8, 31% identity is not "considerably high" (sic), especially when a higher level of similarity is considered "low" a few sentences later!!

RESPONSE. We have corrected the expression by removing “the considerably high”.

l. 139-146, (and figure 2C) computer generated secondary structures have little relevance without corroborating data, I recommend deleting these. They are not put into any context; since these are reported as dsRNA viruses, what is the relevance of the ss structure?

RESPONSE. We have removed the secondary structures and the related information from the manuscript and, when necessary, referred to the coding strand of the dsRNA.

The discussion does not propose any mechanisms for some of the inexplicable points above. All known dsRNA viruses encapsidate a ss pregenome into a virion that contains the RdRp, and the second strand is synthesized in the virion. How could this happen for this virus? Purified viral RNA from dsRNA or even (-) ssRNA viruses has never been found to be infectious, because these viruses require an RdRp to establish infection, they cannot make it directly from their genomic RNA as ssRNA viruses can. How can they explain this? The discussion needs to be expanded to include these points.

RESPONSE. We have dealt with this issue as following: “Normally, a dsRNA virus encapsidates a single stranded pregenome into a virion that contains the RdRp, and the second strand is synthesized within the virion. Here, the lack of association between the RdRp and the CP, as revealed by SDS-PAGE analysis (Fig. 4b), together with CcFV-1 appearing to be intermediate between dsRNA and (+)ssRNA viruses, has led us to conclude that the CcFV-1 genome, in

contrast with typical dsRNA viruses, could be encapsidated following synthesis of the second strand in the host cell. Moreover, its genome may be infectious like a (+)ssRNA. The underlying mechanism needs further studies". We had inserted this conclusion at the end of the first paragraph of the Discussion.

Reviewer #2 (Remarks to the Author):

This reviewer assessed a similar version of the manuscript, submitted earlier to another journal, basically with the same contents by the same authors. The authors incorporated many of my previous suggestions into the current version.

In the paper no NCOMMS-17-01167, the authors report a characterization of a novel fungal virus termed *Colletotrichum camelliae* filamentous virus 1 (CcFV-1). The virus was isolated from a phytopathogenic fungus (strain LT-3-1) that had a 8-segmented dsRNA genome. Findings include: 1) detection of CcFV1 from a fungus unreported as a virus host, 2) hypovirulence conferred by CcFV1 to the fungal host, 3) CcFV1 as an evolutionary intermediate between (+)RNA and dsRNA viruses, 4) the GDNQ motif of CcFV1 RdRp typically possessed by mononegaviruses, 5) extremely long, filamentous particles accommodating CcFV1 dsRNA genomic segments with the dsRNA4-encoded protein, and 6) infectivity of viral dsRNA fractions treated by nuclease S1 and DNase I.

The fact that CcFV1 is closely related to two previously reported fungal viruses: *Aspergillus fumigatus* tetramycovirus 1 (AfuTmV-1) (Kanhayuwa et al., PNAS 112, 2015) and *Botryosphaeria dothidea* virus 1 (BdRV1) (Zhai et al., Virology 493, 2016) lowers the scientific impact of the current paper. Also, recently multiple related viruses were reported by Kotta-Loizou and Coutts (PLoS Pathogen, 2017). Interestingly, they have different genome segment numbers ranging from 5-7 unlike AfuTmV-1 and BdRV1 that have 4 and 5 genomic segments, respectively. Thus the genome segment number 8 of CcFV1 is not as surprising as the authors underline.

As a whole, I feel that CcFV-1's filamentous particle nature and infectivity as dsRNA are of sufficient scientific merit for Nature Communication. However, the first point should be substantiated in a solid way. There follow a few major and minor suggestions:

Major points:

1. As evident from the title of the paper, the most appealing point is the filamentous structure of CcFV1. The authors showed it by ISEM using antibodies to gel-purified 31-kDa protein and confirmed the presence of similar structures in thin sections of infected cells. Given the potential high impact, this notion should be reinforced by another approach. For example, immune-gold labeling of thin sections or immune-trapping of filamentous particles from infected mycelia homogenates, not from purified fractions, should be performed. A related issue is particle length distribution. The authors have provided a sort of histogram (Fig. 3), but the number of tested particles is limited. They should increase them.

RESPONSE. To deal with the requirement of “using immune-gold labeling of thin sections or immune-trapping of filamentous particles from infected mycelia homogenates, not from purified fractions”, we had performed the required experiments with similar results. In fact, we performed the experiments (extracting particles from crude extracts) in the current manuscript according to the previous suggestion “immune-trapping of filamentous particles from infected mycelia homogenates”. Since the background was too dirty to be easily observed with ISEM, we added a further clarification step (at $12,096 \times g$ at $4^{\circ}C$ for 30 min to remove cellular debris) that resulted in clearer visualization. Therefore, there is no essential difference between the results from mycelia homogenates and from the crude extracts that we described. Additionally, we increased the number of tested particles from 20 to 51 (Fig. 3b).

2. No comparison in virion structure is made among the closely related and characterized viruses AfuTmV-1, BdRV1 and CcFV-1. Zhai et al. (Virology 493, 2016) reported “bacilliform virus-like particles about 30–80 nm in length and 13.4 nm in diameter” as presumable BdRV1 virions, although they failed to prove it. Kanhayuwa et al. (PNAS 112, 2015) showed the unusual AfuTmV-1 structure as an infectious entity associated with P(proline)-A(alanine)-S(serine)-rich protein in a colloidal form. Therefore, there is a huge discrepancy in virion morphology among these closely related “tetraviruses.” The authors should mention this in an appropriate place. Also I would suggest the authors to test the purification methods of Zai et al. and Kanhayuwa et al. for CcFV1 and state their results.

RESPONSE. Yes, we have discussed the discrepancy in virion morphology in the revised version in page 16.

We have compared our protocol and that of Zai et al., and found that both produced similar results. The particles (“bacilliform virus-like particles about 30-80 nm in length and 13.4 nm in diameter”) reported by Zhai et al. (Virology 493, 2016) appear similar (although shorter) to those observed in our study, perhaps as a consequence of the extracting process. We observed a situation of this kind at the beginning of our studies when we crushed mycelia using a juice squeezer. Since the protocols that we have used produced a clear result, we do not think necessary to apply the protocols of Kanhayuwa et al.

3. Kotta-Loizou and Coutts recently reported viruses with 4-7 dsRNA genomic segments closely related to AfuTmV1 (PLoS Pathogens). A thorough comparison should be made and briefly mentioned in an appropriate place.

RESPONSE. Since the genomic sequences are not freely accessible in NCBI for BbPmV-1, -2 and -3, there are no blast results about their identities between CcFV-1 and BbPmV-1, -2 and -3. However, we asked the sequences from Kotta-Loizou, and made a thorough comparison between the new reported viruses with CcFV-1.

4. Perhaps I missed it, but did the authors test “filamentous” particles for infectivity? The authors should touch whatever its outcome may be.

RESPONSE. Since numerous reports had been described that fungal mycelia can be easily transfected with viral particles, it is beyond doubt that the *Camellia sinensis* can be easily transfected by CcFV-1, producing new virions. However, transfected with purified particles, it raised the doubt that the particles might be originated from other contaminated filamentous virus particles or from host proteins, because it is difficult to get absolutely pure virions for transfection. Therefore, infectivity with “filamentous” particles had no substantive support to the major conclusion, and does not yield a meaningful outcome. Therefore, we conducted infectivity with naked dsRNAs here, providing a more strict proof.

4. Condense pages 6-9.

RESPONSE. Yes we tried to condense the paragraphs by removing the secondary structures and other less important information.

Reviewers' comments:

Reviewer #1 (Remarks to the Author):

Most of the issues raised in the first review have not been addressed in this version of the paper. There is still no convincing data to show that the RNAs are truly double-stranded, and even if the authors state in their rebuttal letter that they did RNase digestion this has to be described in the PAPER, along with the method used to extract "dsRNA".

I do not find the paper acceptable as modified, but encourage the authors to do the appropriate experiments to confirm the dsRNA nature of the genome, which still seems highly doubtful, given that there is no known mechanism for naked dsRNA to be infectious. This requires a number of things that defy the current understanding of molecular biology. If they are in fact true this needs to be shown beyond any doubt.

Reviewer #2 (Remarks to the Author):

The authors addressed some of my last concerns, but points 2 and 4 were not addressed appropriately.

Point 2. Coutts and colleagues purified a colloidal form of virions for a similar virus, AfuTmV-1 by a method CsCL. The authors should attempt their method for Colletotrichum camelliae filamentous virus 1 (CcFV-1) to see if a similar fraction is obtained. This reviewer is not saying that the authors should observe virions by atomic force microscopy.

Point 4. The infectivity, though at a low level, of CcFV-1 as dsRNA is of great interest. However, as this virus appears to form filamentous particles, it would be interesting to determine whether the unusual filamentous particles can infect host protoplasts more efficiently than its dsRNA. The authors' rebuttal to this issue is not convincing at all. They could semi-purify particles and presented their EM pictures.

I would recommend that the authors perform the suggested experiments which are not difficult.

Response to the reviewers

Reviewer #1 (Remarks to the Author):

Most of the issues raised in the first review have not been addressed in this version of the paper. There is still no convincing data to show that the RNAs are truly double-stranded, and even if the authors state in their rebuttal letter that they did RNase digestion this has to be described in the PAPER, along with the method used to extract "dsRNA".

I do not find the paper acceptable as modified, but encourage the authors to do the appropriate experiments to confirm the dsRNA nature of the genome, which still seems highly doubtful, given that there is no known mechanism for naked dsRNA to be infectious. This requires a number of things that defy the current understanding of molecular biology. If they are in fact true this needs to be shown beyond any doubt.

Response: We conducted two independent experiments to prove the dsRNA nature of CcFV-1 genomic components.

1) The dsRNA nature of the eight observed bands was assessed by treatments with RNase III, S1 nuclease or RNase A (in 2× and 0.1×SSC), together with an ssRNA control [*in vitro* dimeric transcripts of citrus exocortis viroid (CEVd)] and dsRNAs from a dsRNA mycovirus (*Botryosphaeria dothidea chrysovirus* 1, BdCV 1). The RNAs extracted from strain LT-3-1 together with BdCV 1 dsRNAs were digested into approximately 20 bp-sized fragments by RNase III and degraded by RNase A in 0.1×SSC, but they resisted digestion by S1 nuclease and RNase A in 2×SSC. In sharp contrast, CEVd transcripts were completely degraded by S1 nuclease and by RNase A under both ionic conditions, but resisted digestion by RNase III (Fig. 1b). These data strongly support that the RNAs extracted from strain LT-3-1 were indeed dsRNAs.

The results were involved in Fig. 1b and lines 95-105 of pages 5 and 6 in the modified manuscript, and the figure was also indicated as follows (Fig. 1).

Fig. 1 Electrophoresis analysis of enzyme-treated nucleic acid samples on 1.2% agarose gels. Samples were treated with RNase III (I), S1 nuclease (II) and RNase A (in 2 \times and 0.1 \times SSC) (III), respectively. “-” and “+” refer to incubated in the reaction buffer without and with the enzyme, respectively. CEVd, ssRNA transcripts (approximately 750 nt) from dimeric cDNAs of citrus exocortis viroid (CEVd). The upper band on the lane of CEVd sample correspond to the remnant plasmid used for transcription, and the lower intense band to the transcript.

2) We successfully cloned the full-length dsRNAs 7 and 8 by ligating a 3'-closed adaptor and RT-PCR amplification using only primers complementary to the adaptor; a ssRNA cannot be amplified using this strategy.

Reviewer #2 (Remarks to the Author):

The authors addressed some of my last concerns, but points 2 and 4 were not addressed appropriately.

Point 2. Coutts and colleagues purified a colloidal form of virions for a similar virus, AfuTmV-1 by a method CsCl. The authors should attempt their method for Colletotrichum camelliae filamentous virus 1 (CcFV-1) to see if a similar fraction is obtained. This reviewer is not saying that the authors should observe virions by atomic force microscopy.

Response: We had conducted virion purification using centrifugation through a CsCl cushion. The procedures were carried out according to the protocol described by Coutts and colleagues (*Proc. Natl Acad. Sci., USA* 112, 9100-9105, 2015) with a minor modification: using PB (8.0 mM Na₂HPO₄, 2.0 mM NaH₂PO₄, pH 7.0) rather than TE as the initial buffer for resuspension of the ground mycelium powder. The results showed that filamentous viral particles, together with their dsRNAs and coat proteins, were successfully extracted from LT-3-1 strain, with the longest one having a width of 16.3 nm and a length of 3266.6 nm (Fig. X4, see below), but not from the virus-lacking strain LT-3-1D2.

The purified particles are shown in the supplementary Fig. 3c in the new submitted manuscript, and the related figures were also indicated as follows (Fig. 2).

Fig. 2. dsRNAs, viral coat protein, and virus-like particles extracted from *C. camelliae* strain LT-3-1 according to the protocol described by Coutts and colleagues (*Proc. Natl. Acad. Sci., USA* 112, 9100-9105, 2015).

Point 4. The infectivity, though at a low level, of CcFV-1 as dsRNA is of great interest. However, as this virus appears to form filamentous particles, it would be interesting to determine whether the unusual filamentous particles can infect host protoplasts more efficiently than its dsRNA. The authors' rebuttal to this issue is not convincing at all. They could semi-purify particles and presented their EM pictures. I would recommend that the authors perform the suggested experiments which are not difficult.

Response: We conducted the transfection of strains LT-3-1D2 and DP-3-1 using purified particles (70-80 μg) in parallel with CcFV-1 dsRNAs (5-7 μg). Our results showed that LT-3-1D2 and DP-3-1 protoplasts were successfully transfected by dsRNAs although in a low efficiency (3/129 and 10/94 of colonies, respectively), but not by particles (0/80 colonies), possibly because most of the particles were fragmented during the isolation/purification procedures.

These results are presented and discussed in lines 319-327 of page 16 in the modified manuscript.

REVIEWERS' COMMENTS:

Reviewer #2 (Remarks to the Author):

This reviewer has been almost satisfied with the current version of the manuscript submitted to Nature Communications. This reviewer has no major reservation, but the lack of infectivity of purified CcFV-1 filamentous particles is a bit surprising. The authors assumed it to be due to fragmentation of virus particles (see rebuttal). However, Fig. 5 shows no trace of fragmentation of filamentous particles, i.e., genomic segments. There may be some other reasons which warrant further exploration in the future.

Response to the reviewer:

REVIEWERS' COMMENTS:

Reviewer #2 (Remarks to the Author):

This reviewer has been almost satisfied with the current version of the manuscript submitted to Nature Communications. This reviewer has no major reservation, but the lack of infectivity of purified CcFV-1 filamentous particles is a bit surprising. The authors assumed it to be due to fragmentation of virus particles (see rebuttal). However, Fig. 5 shows no trace of fragmentation of filamentous particles, i.e., genomic segments. There may be some other reasons which warrant further exploration in the future.

RESPONSE: Numerous reports have described successful transfection with isometric particles to the fungal host cells. However, little is known about using filamentous particles for a successful transfection, which might be due to that filamentous particles are easily broken in the purified process. We will put in effort to get insight into the underlying mechanism in further studies.